# Knee and Peri-Knee Tissues of Post Mortem Donors Are Strategic Sources of Mesenchymal Stem/Stromal Cells for Regenerative Procedures

**DOI:** 10.3390/ijms23063170

**Published:** 2022-03-15

**Authors:** Gregor Haring, Janja Zupan

**Affiliations:** 1Institute of Forensic Medicine, Faculty of Medicine, University of Ljubljana, SI-1000 Ljubljana, Slovenia; gregor.haring@mf.uni-lj.si; 2Department of Clinical Biochemistry, Faculty of Pharmacy, University of Ljubljana, SI-1000 Ljubljana, Slovenia

**Keywords:** mesenchymal stem/stromal cells, post mortem donors, knee, peri-knee tissues, bone and bone marrow, synovium, periosteum, in vitro analysis, human skeletal stem cell markers

## Abstract

Tissues of post mortem donors represent valuable alternative sources for the isolation of primary cells with mesenchymal stem/stromal cell (MSC)-like properties. However, the properties of primary cells derived from different tissues and at different post mortem times are poorly recognized. Here, we aim to identify the optimal tissue source between three knee and peri-knee tissues for the isolation of primary cells with MSC-like properties, and to define the influence of the time post mortem on the properties of these cells. We harvested tissues from subchondral bone marrow, synovium and periosteum from 32 donors at various post mortem times. Primary cells were evaluated using detailed in vitro analyses, including colony formation, trilineage differentiation, immunophenotyping and skeletal stem cell marker-gene expression profiling. These data show that the primary cells with MSC-like properties isolated from these three tissues show no differences in their properties, except for higher expression of *CD146* in bone-marrow cells. The success rate of the primary cell isolation is dependent on the post mortem time. However, synovium and periosteum cells isolated more than 48 h post mortem show improved osteogenic and chondrogenic potential. This study suggests that knee and peri-knee tissues from donors even 3 days post mortem are strategic sources of MSCs for regenerative procedures.

## 1. Introduction

Mesenchymal stem/stromal cells (MSCs) and their derivatives (i.e., exosomes) represent major tools in regenerative medicine. The regenerative and immunomodulatory properties of MSCs were demonstrated decades ago, in particular for treating degenerative joint disorders such as osteoarthritis [1,2]. Moreover, tissue-specific subsets of MSCs even show joint morphogenic properties, as shown in animal joint injury models [3]. In humans, the use of autologous or allogeneic cells that are most commonly derived from bone marrow or adipose tissue has reshaped the treatment of debilitating age-associated or injury-inflicted joint disorders. However, there remains substantial concern for the use of autologous cell therapies in patients who already have degenerative disorders, as the regenerative potential of the endogenous MSCs might be impaired or exhausted [4,5,6,7,8]. Degenerative comorbidities and the increasing age of the donor might also have an impact on the quantity of obtainable MSCs [6]. 

For autologous cell therapies, the quantities of these cells are also largely limited by the amount of biopsy tissue that can be harvested from a patient. Culture-expanded cells offer a solution to overcome the limited amounts of primary cells that can be isolated from patient tissues; however, in vitro expansion might also result in significant losses of the regenerative and immunomodulatory properties of the MSCs obtained. Moreover, there is also the problem of the heterogeneity of MSCs included in such cell therapies. Functional MSC heterogeneity exists among donors, tissues, and MSC subpopulations, which results in differences in their potential, in particular for cartilage repair [9]. 

For more than a decade, the most commonly used criteria to identify the MSCs in vitro were those defined by the International Society for Cellular Therapy (ISCT) [10]. These represent the set of three minimal criteria: (i) MSCs must be plastic-adherent when maintained under standard culture conditions; (ii) MSCs must express CD105, CD73, and CD90, and lack expression of CD45, CD34, CD14 or CD11b, CD79a or CD19, and HLA-DR surface molecules; and (iii) MSCs must differentiate to osteoblasts, adipocytes, and chondroblasts in vitro [10]. With unfolding of new knowledge in the field of MSCs, it has become clear that the MSCs defined by these relatively loose criteria present large heterogenous groups of stem cells and that their characteristics do not reflect their in vivo function [11,12].

Great advances in the identification of specific subpopulations of MSCs with clearly defined properties both in vitro and in vivo have been made in recent years, with the identification of the phenotypes of human skeletal stem cells (SSCs) as podoplanin (PDPN)/CD164/CD73 positive and CD146 negative [13]. The studies that followed provided further evidence that ageing is associated with progressive loss of SSCs and diminished chondrogenesis in the joints, which appears to lead to joint degeneration and osteoarthritis [14]. Moreover, Murphy et al. (2020) also provide a solution as to how the chondrogenic potential of SSCs can be boosted, i.e., with a combination of microfracture surgery to the chondral surface of adult joints, followed by the localised co-delivery of bone morphogenetic protein 2 (BMP2) and a soluble vascular endothelial growth factor receptor (sVEGFR)1 antagonist [14]. The well-defined properties and immunophenotype of the SSCs identified a subpopulation of stem cells with clearly defined stem cell regulation and hierarchical organization of human skeletal progenitors, as opposed to vaguely defined MSCs that can be virtually obtained from any connective tissue on the basis of the ISCT criteria [10].

The tissues harvested from post mortem donors represent an alternative tissue source for isolating MSCs, particularly with the availability and ease of access to large numbers of biopsies [15,16]. Studies have shown that primary cells from various tissues of cadaveric donors can be isolated several hours, or even several days, post mortem [15]. Reassuringly, studies that have compared the properties of MSCs derived from cadaveric donors with those from living donors have shown comparable results [17,18]. However, face-to-face comparisons of the effects of various times post mortem on the properties of primary MSC-like cells derived from different tissues are scarce [15]. 

The main aim of the present study was to perform a comprehensive in vitro analysis of primary MSC-like cells derived from three knee and peri-knee tissues, to determine whether the properties of these cells differ between these tissues and for the times post mortem. Following recent studies [13,14], we also verified the expression of the markers of humans SSCs. The present study shows that primary cells from all three of these tissues are comparable in their in vitro MSC-like characteristics. These data also show that the time post mortem affects the in vitro characteristics of MSCs derived from a given tissue. Interestingly, longer times post mortem are not always associated with decreased in vitro performance of these MSCs. Finally, the isolated MSCs from these tissues expressed genes that code for SSCs, where the synovium and periosteum tissues show profiles more resembling SSCs than bone tissue.

Overall, the present study underpins the usefulness of the readily available post mortem tissues as valuable sources of MSC-like cells for regenerative medicine. These data warrant further studies using these MSCs in particular as sources of SSCs for chondral lesion regeneration, as suggested by recent studies [14].

## 2. Results

### 2.1. Donors and Tissues

Tissues from 32 donors were included in this study. The basic characteristics, causes of death, and times post mortem of the donors before the tissues were harvested are given in Table 1. 

The donors were classified into three groups according to the times post mortem before the tissues were harvested: ≤24 h (range, 4–24 h); 24 to 48 h (range, 25–45 h); and ≥48 h (range, 52–108 h). The means of the characteristics for each time post mortem group are given in Table 2. The time post mortem groups did not differ in mean age or body mass index (*p* > 0.05; one-way ANOVA with Bonferroni multiple comparisons tests), or for the male/female ratio of the donors (*p* > 0.05; Chi-squared tests).

All of the three tissues, as bone and bone marrow (BM), synovium, and periosteum, were harvested at the knee and peri-knee sites as shown in Figure 1a. In total, 96 biopsies (Figure 1b) were collected for primary cell isolation. The distributions of the tissue weights obtained are shown in Figure 1c, where significantly greater BM biopsy weights (mean, 0.568 g) were obtained compared to synovium and periosteum (means, 0.280, 0.113 g, respectively; *p* < 0.0001 for both), and significantly greater weights were obtained for synovium compared to periosteum (*p* < 0.05; one-way ANOVA with Bonferroni multiple comparisons tests).

### 2.2. Bone Marrow Shows the Highest Primary Cell Isolation Efficiency

The data for the successfully isolated and culture-expanded primary cells (calculated as a percentage of donors per time post mortem group shown in Table 2) from each tissue according to the time post mortem groups and the number of samples for the in vitro analyses are given in Table 3.

Here, the BM tissue provided the greatest proportion of successfully culture-expanded primary cells (Figure 2a). For the BM tissue, the cells grew in vitro in 26 of 32 samples, whereas for the synovium and periosteum tissues, the cells grew in vitro in 19 and 16 of 32 samples, respectively. Most frequently, the lower success rates of isolation in the synovium and periosteum were due to bacterial or fungal infections or due to lack of adherent cells. These two tissues also required longer collagenase digestion (i.e., 12 h), compared to BM tissue (3 h). Across the time post mortem groups, the lowest rate of successful isolation was for the group with time post mortem ≥ 48 h for all tissues, as was expected (Table 2; Figure 2b).

### 2.3. Bone Marrow Cells Isolated 24 to 48 h Post Mortem Show the Slowest Colony Formation

The successfully culture-established primary cells did not show any significant differences in time to first observation of plastic adherence of the cells (Figure 3a,b). There were also no differences in time to colony forming unit fibroblast assay (CFU-F) at passage (p) 0 between the tissues (Figure 3c). However, the time post mortem groups comparison here (Figure 3d) shows that bone marrow (BM) cells isolated 24 to 48 h post mortem required significantly longer time to CFU-F compared to their counterparts isolated at both ≤24 h and ≥48 h post mortem (*p* < 0.0001, *p* < 0.05, respectively; two-way ANOVA with Bonferroni multiple comparison tests). The primary cells from all of the tissues and time post mortem groups showed similar colony establishment, as demonstrated by the CFU-F p0 (Figure 3e–g). 

### 2.4. Synovium Cells Isolated ≥ 48 h Post Mortem Show Significantly Higher Osteogenesis Compared to Their Counterparts Isolated 24 to 48 h Post Mortem

Comparisons of the osteogenic potentials as determined by Alizarin red S concentrations did not show any differences between the tissues (Figure 4a,c). For the osteogenic potential between the time post mortem groups (Figure 4b), primary cells isolated from synovium ≥ 48 h post mortem showed significantly higher Alizarin red S concentrations compared to those isolated from the same tissue between 24 and 48 h post mortem (*p* < 0.05; two-way ANOVA with Bonferroni multiple comparison tests). Comparisons of the expression of osteogenesis-related genes showed no differences between the tissues or between the time post mortem groups (Figure 4d,e).

### 2.5. Adipogenic Potential of Primary Cells Is Similar between the Tissues and Time Post Mortem Groups

Comparisons of the adipogenic potentials, as determined by the proportions of Oil red O positive adipocytes, did not show any significant differences between the tissues (Figure 5a,c) or between the time post mortem groups (Figure 5b), and the same occurred for the expression of the selected adipogenesis-related genes (Figure 5d,e).

### 2.6. Periosteum Cells Isolated ≥ 48 h Post Mortem Show Significantly Higher Chondrogenesis Compared to Their Counterparts Isolated 24 to 48 h Post Mortem

No differences were seen between the tissues for comparison of the chondrogenic potentials, as determined by visual evaluation for the quality assessment of in vitro engineered hyaline-like cartilaginous tissue (i.e., Bern score) (Figure 6a), and for chondrogenic pellet diameter (Figure 6c,e). However, for the chondrogenic potential across the time post mortem groups, periosteum cells isolated ≥ 48 h post mortem showed higher Bern scores (Figure 6b) compared to their counterparts isolated 24 to 48 h post mortem (*p* < 0.05; two-way ANOVA with Bonferroni multiple comparison tests). Synovium cells isolated 24 to 48 h post mortem formed chondrogenic pellets with the greatest diameters (Figure 6d), compared to the ≤24 h and ≥48 h time post mortem groups (*p* < 0.05, *p* < 0.01, respectively; two-way ANOVA with Bonferroni multiple comparison tests). 

The chondrogenic potentials of the primary cells were also determined through immunofluorescence for the α-1 chain of type II collagen (Col2A1), as an indication of hyaline cartilage. There were no differences in the frequencies of Col2A1 positivity between the tissues (Figure 7a,c) or between the time post mortem groups (Figure 7b) (*p* > 0.05; Chi-squared tests).

### 2.7. Bone Marrow Cells Isolated ≥ 48 h Post Mortem Show Lower Expression of the MSC Marker CD90

The quality of the isolated and culture-expanded primary cells was confirmed according to the criteria for MSCs suggested by the International Society for Cellular Therapy (ISCT) [10], through immunophenotyping of these primary cells from p1 to p5. Comparisons of the expression of the positive markers CD73, CD90, and C015 (Figure 8a) showed no differences between the tissues (*p* > 0.05; two-way ANOVA with Bonferroni multiple comparison tests). For the time post mortem groups (Figure 8b), BM cells isolated ≥ 48 h post mortem showed significantly lower expression of CD90, a positive marker of MSCs, compared to their counterparts isolated ≤24 h and 24 h to 48 h post mortem (*p* < 0.05, *p* < 0.01, respectively; two-way ANOVA with Bonferroni multiple comparison tests). Moreover, synovium cells isolated 24 to 48 h post mortem showed significantly higher expression of CD105, a positive MSC marker, compared to their counterparts isolated ≥ 48 h post mortem (*p* < 0.01; two-way ANOVA with Bonferroni multiple comparison tests). There were no differences in the expression of the combination of the CD45/CD19/CD14 negative MSC markers between the tissues (Figure 8c) or between the time post mortem groups (Figure 8d). The primary cells in all three of the tissue and time post mortem groups showed a mean expression of the negative markers as <2%, as suggested by the ISCT (Figure 8c,d), whereas the mean values for all three of the positive markers did not reach the criteria of 95% for any of the tissues or time post mortem groups (Figure 8a,b).

### 2.8. Bone Marrow Cells Show Significantly Higher Expression of the Negative SSC Marker CD146 Compared to Synovium and Periosteum Cells

Gene expression profiling for the expression of genes recently identified as markers of human SSCs (i.e., PDPN, CD73, CD164, CD146) [13] was carried out for the in vitro cultured cells between p1 and p3. Interestingly, the BM cells showed significantly higher expression of the negative marker CD146 compared to the synovium and periosteum cells (Figure 9a) (*p* < 0.01 for both; two-way ANOVA with Bonferroni multiple comparison tests). For the comparison of the expression profiles of SSC markers between the time post mortem groups (Figure 9b), there were no significant differences seen (*p* > 0.05; two-way ANOVA with Bonferroni multiple comparison tests). 

Hierarchical clustering of SSC marker-gene expression was also carried out, as illustrated in the heat map in Figure 9c. Here, two clusters were identified (Figure 9c, blue circles) with higher expression of the negative marker CD146 and lower expression of the positive markers PDPN, CD73, and CD164. These encompass mainly BM-derived cells (7/10, 10/14 samples). Clustering tree analysis was also carried out (Figure 9c, left).

## 3. Discussion

Mesenchymal stem/stromal cells are vital components of cellular therapies, as they show immense potential for treating various degenerative disorders, and in particular for joint degeneration in osteoarthritis [19]. This debilitating disorder of the movable joints is on the rise with the ageing of the world population [20], with the currently available treatment options limited to major surgical interventions, such as arthroplasty. Primary cells with MSC-like properties are routinely obtained by well-established isolation and culture-expansion protocols. However, their quantities and regenerative capacities are dependent on the age and pathophysiological condition of the donor [1,8]. In particular, the concomitant presence of already established osteoarthritis might affect the endogenous regenerative potential of the primary cells [5,7,21], and thus question the efficacy of such cell therapies, in particular when autologous.

Post mortem tissues represent plentiful alternative sources for such cellular therapies, in particular from young donors without signs of age-associated joint degeneration. Previous studies have shown that primary cells with MSC-like properties can be obtained from various tissues, even several days post mortem [15,16,18,22,23,24], and from tissues stored under cryoprotection for years [25]. However, comprehensive studies that include several tissues from human post mortem donors along with specific times post mortem are scarce. 

In the present study we focused on three knee and peri-knee tissues from 32 post mortem donors (i.e., 96 tissue biopsies collected in total), as these are more straightforwardly accessible during routine autopsies, as illustrated in Figure 1. Bone marrow is the most commonly used source of MSCs. As a previous study showed that cells released enzymatically from trabecular bone are virtually identical to BM-aspirate-derived MSCs [26], we used tibial subchondral trabecular bone with bone marrow as a source of MSCs here. On the other hand, synovium was used based on previous reports that have shown the particular chondrogenic and joint morphogenetic properties of the MSCs derived from this tissue [3,17,27]. Further, periosteum was chosen as a less well-recognized tissue, although based on previous reports, it still represents a promising tissue source of MSC-like cells [18,22]. 

In addition to comparisons of these three tissues in terms of their potential for isolating MSC-like cells in vitro, we also compared the three groups of donors according to the times post mortem within each tissue, before the tissues were harvested: ≤24 h; 24 to 48 h; ≥48 h. There were no differences between the time post mortem groups for age, sex ratio, and body mass index, which are well-recognized risk factors for osteoarthritis [28]. Here, these three tissues were available for harvesting in greater quantities than for living donors, with the weight of the harvested biopsies highest for the tibial subchondral BM and synovium, compared to periosteum. The periosteum had the lowest amounts available due to the more complex tissue harvesting process, as this tissue needs to be separated from the underlying cortical bone and cleaned of muscle and any other soft tissue attached to its surface.

The present study identified tibial subchondral BM as the most reliable source for the isolation and establishment of primary cells with MSC-like properties, across all three of the time post mortem groups. The success rates for the isolation of the primary cells were lower for synovium and periosteum, which might be due to the longer processing times (which included 12 h collagenase digestion) and the higher rates of bacterial and fungal infections in the p0 cultures.

As expected, the success rate of primary cell isolation decreased gradually for all three of the tissues with increasing time post mortem. Indeed, the tissues harvested from the donors with the two longest times post mortem here (i.e., 108, 101 h) did not produce any cells. Further culture expansion of all of the primary cells after p0 was also required to obtain sufficient quantities of cells for all the in vitro analyses. Across all of the time post mortem groups, BM was the tissue that provided the highest number of primary cell samples for inclusion in these analyses. 

The detailed in vitro analyses showed no differences between the tissues studied in terms of time to plastic adherence of the primary cells, time to colony formation, formation of colonies (CFU-F), osteogenic, adipogenic, and chondrogenic potentials at the gene and protein levels, and immunophenotypes as recommended by the ISCT [10]. However, there were differences in gene expression profiling for the recently identified markers of human SSCs, namely, *PDPN*, *CD73*, *CD164*, and *CD146* [13]. Interestingly, BM cells showed the highest gene expression of the negative marker *CD146* in comparison with synovium and periosteum cells. Moreover, heat map analysis identified two clusters of cells with high *CD146* expression and low *PDPN*/*CD73*/*CD164* expression, where the majority of these primary cells were from BM. These data suggest that tibial subchondral trabecular bone of post mortem donors might not be the optimal source of SSCs, as other tissues such as synovium and periosteum have gene expression profiles more closely resembling that of SSCs. However, the data need to be further confirmed, first at the protein level, and secondly with functional analyses of the sorted cells isolated from these tissues. 

Furthermore, to identify the influence of time post mortem on their properties, the present study compared the MSC-like in vitro properties within each of the tissues among the three time post mortem groups. Significant differences between the time post mortem groups were found for BM in terms of the time needed for these primary cells to establish colonies. Interestingly, the BM cells isolated 24 to 48 h post mortem required the longest time to form colonies compared with the other two time post mortem groups, although this was to be expected for the longest time post mortem group.

Significant differences were also shown between the time post mortem groups for osteogenic potential. Surprisingly, primary cells isolated from synovium at ≥48 h post mortem showed significantly higher osteogenic potential in comparison to those isolated from synovium 24 to 48 h post mortem. Similarly, as assessed by the Bern scores, periosteum cells isolated ≥ 48 h post mortem showed higher chondrogenic potential compared to their counterparts isolated 24 to 48 h post mortem. Synovium cells isolated 24 to 48 h post mortem formed the largest chondrogenic pellets, compared to the other two time post mortem groups. However, this property does not reflect the quality of the chondrogenic matrix, as assessed by the Bern score [29]. Although the primary cells in the present study showed high osteogenic and chondrogenic potential even 3 days post mortem, the post mortem time accounts for substantial changes in the tissue microenvironment in relation to blood and oxygen supply. The MSCs are therefore subjected to inflammatory triggers and reactive oxygen species (ROS) formation. Previous studies have shown that ROS affect the properties of MSCs via degradation of the extracellular matrix, and in particular the collagen fibres, which shifts MSC differentiation toward adipocytes instead of osteoblasts [30]. Moreover, the microenvironmental pH is one of the crucial parameters that greatly affects tissue repair and homeostasis. Interestingly, Hazehara-Kunitomo et al. showed that the acidic pH during the initial stages of bone healing is important to enhance the stem cell properties of BM-derived MSCs [31]. Similarly, Massa et al. showed that an acidic microenvironment promotes the maintenance of stemness of osteogenic MSCs through induction of stemness-related genes and a quiescent cell-cycle status [32]. As acidosis is characteristic of the post mortem tissue microenvironment, this might also explain the higher osteogenic potential of the synovium cells with longer times post mortem found in the present study. On the other hand, chronic metabolic acidosis alters osteoblast differentiation from MSCs, which results in impairment of bone formation, as shown by Disthabanchong et al. [33].

Collagen is a major component of the extracellular matrix of the connective tissue, and it has a role in their homeostasis and in wound healing [34]. It has been suggested that tissue trauma can lead to collagen proteolysis, which exposes a host of cell and ligand binding sites that are crucial for tissue regeneration [34]. However, the present study did not show any differences in collagen type I expression in osteogenesis, or in collagen type II synthesis in chondrogenesis. This might be because the MSCs analysed in the present study were derived from the knee and peri-knee tissues, where minimal or no trauma had been inflicted.

Moreover, significant differences were obtained for the immunophenotypes of these primary cells, as suggested by the ISCT [10]. The ISCT guidelines indicate that MSC-like cells must express <2% of the negative markers (i.e., CD45, CD19, CD14) and >95% of the positive markers (i.e., CD73, CD90, CD105) [10]. The mean expression of the negative markers reached these criteria for all of the tissues and time post mortem groups, whereas the mean expression of the positive markers did not reach these criteria in any of these groups. In particular, lower expression of CD105 was observed in all of the tissues and time post mortem groups. This feature was frequently observed in our previous studies on primary MSCs from living donors [4,5,35,36] as also in other studies with MSC-based clinical trial products [37], which questions the usefulness of CD105 as a marker for MSCs. However, in the present study, synovium cells isolated 24 to 48 h post mortem expressed significantly higher proportions of CD105 compared to their counterparts isolated ≥ 48 h post mortem. This study also showed that in BM cells, longer time post mortem affected the expression of CD90. The BM cells isolated ≥ 48 h post mortem showed significantly lower expression of CD90 compared to their counterparts isolated ≤ 24 h post mortem and 24 to 48 h post mortem. For the rest of the parameters, which included time to plastic adherence, colony formation, adipogenic potential, and gene expression of the SSC markers, there were no significant differences between the time post mortem groups.

In the present study we followed the commonly used ISCT criteria to show that primary cells derived from post mortem knee and peri-knee tissues have MSC-like properties in vitro, and the criteria allowed us to compare them to similar MSC populations obtained from living donors. As the ISCT criteria define very heterogenous populations of MSCs and indicate only their in vitro properties, we wanted to take a step further and verify the expression of the markers of human SSCs. Identification of human SSCs [13] represents a big step forward for better identification of MSCs after more than a decade of using the markers suggested by the ISCT [10]. In addition to their identity and their generation of progenitors of bone, cartilage, and stroma, an approach on how to boost the chondrogenic potential of SSCs for cartilage regeneration has also been demonstrated [14]. The results of the gene expression profiling for SSC markers in the present study indicate that the synovium and periosteum primary cells resemble the SSC profile more closely than BM cells. However, the synovium and periosteum primary cells in the present study did not show significantly higher potential for osteogenesis or chondrogenesis in comparison with BM cells. Nevertheless, the chondrogenic potential of synovium-derived as well as periosteum-derived MSCs is well-recognized from previous studies [3,17,18]. Given that our study shows that synovium and periosteum primary cells resemble the SSC profile more closely than BM cells, these cells should be further investigated for their potential as cellular sources for chondral lesion regeneration. 

There are also limitations to the present study. The major limitation is that it was not possible to perform all of the analyses for the complete study cohort, i.e., for all of the donors included and also for all of the successfully culture-expanded samples, as shown in Table 3. Of note, the inclusion of donors in this study over the last 2 years slowed down substantially due to the SARS-CoV-2 epidemic, where only donors who tested negative for this virus could be included. Isolation of primary cells is also a relatively long procedure, and our previous studies have shown that the success of the isolation procedure can differ greatly between living donors [4,5,35]. Some cells did not expand enough beyond p0 for the multiple analyses to be performed here, as these required substantial numbers of cells. However, many of these limitations also apply to other studies, and the size of the present study cohort was comparable to other studies [7,17,18,21,22]. Instead, the advantage of the present study is the comprehensive approach to the analysis of primary MSC-like cells of several knee and peri-knee tissues that are more readily available and easily accessible (in comparison with living donors) and that were harvested at various times post mortem. An appealing alternative to overcome the hurdles of primary MSC isolation and their tedious culture expansion is the use of their derivatives, and in particular, of their exosomes [38]. The exosomes secreted by MSCs carry the whole precious biological information that characterises the use of MSCs in regenerative medicine, and at the same time, they allow for more easy and available storage and distribution [38].

To summarise, the present study indicates that all three of these knee and peri-knee tissues, as BM, synovium, and periosteum, represent valuable sources of MSC-like cells, with no major differences in their in vitro properties except for the higher expression of *CD146* in BM cells. Time post mortem affected the success rate of the isolation and culture expansion of the primary cells from all three of the tissues. However, longer times post mortem still resulted in the production of primary cells with MSC-like in vitro properties, as seen here for up to 69 h. Furthermore, some of the in vitro properties were improved with longer time post mortem, such as the osteogenic potential of synovium cells and the chondrogenic potential of periosteum cells.

These data underpin the findings from previous studies on the usefulness of post mortem tissues in regenerative medicine. They also warrant further study, in particular on MSCs derived from synovium and periosteum of post mortem donors in terms of chondral lesion regeneration, as suggested by other recent studies.

## 4. Materials and Methods

### 4.1. Donor Inclusion and Tissue Harvesting

Donors were included and their tissues harvested at the Institute of Forensic Medicine, Faculty of Medicine, University of Ljubljana (Slovenia). Approval for this study was obtained from the National Medical Ethics Committee of the Republic of Slovenia (reference numbers: 0120-523/2016/15, date of approval 16/05/2018). Knee (synovium) and peri-knee (subchondral bone with bone marrow, periosteum) tissues were harvested from each donor during routine autopsies at different times post mortem. 

The subchondral bone and periosteum tissue (both approximately 1 cm^3^ in size) were harvested following previous studies [18,22]. More specifically, they were taken from the proximal medial tibia, medial from the line “intercondylar eminence–tibial tuberosity”, and below the medial condyle and synovium (approximately 1 cm^3^ in size) in the area under the medial or lateral collateral ligament. All of the tissues harvested were stored in low glucose Dulbecco’s modified Eagle’s medium (DMEM; Biowest, Nuaillé, France) supplemented with 10% foetal bovine serum (Gibco, Thermo Fisher Scientific, Waltham, MA, USA), 1% glutamine, 2% penicillin, and streptomycin (all Biowest), until cell isolation.

### 4.2. Cell Isolation 

Primary cells from the three tissues were isolated at the Faculty of Pharmacy, University of Ljubljana. The isolation protocols followed previously published studies for isolating primary cells from bone tissue [5,26], synovium [17,39], and periosteum [18,22]. Briefly, the tissues were cut into small pieces, washed thoroughly in phosphate-buffered saline, weighed, and incubated at 37 °C in 1 mg/ mL collagenase solution (Roche, Basel, Switzerland) for 3 h (bone tissue) or 12 h (synovium and periosteum). The resulting suspensions of tissue and cells was passed through a 70 µm cell strainer (Corning Inc., Corning, NY, USA). Aliquots of freshly isolated cells were seeded using StemMACS MSC expansion media kit XF, human (Miltenyi Biotec, Bergisch Gladbach, North Rhine-Westphalia, Germany) supplemented with 1% glutamine, 2% penicillin, and streptomycin (all Biowest). The cells were incubated at 37 °C under 5% humidified CO_2_. The study design and the analyses are summarized in Figure 10. 

### 4.3. Colony Forming Unit Fibroblast Assay and Culture Expansion

The colony forming unit fibroblast assays (CFU-F) were performed as described previously [39]. Briefly, freshly isolated cells were plated at p0 as nine replicates in six-well plates. Once the colonies were formed, six wells were trypsinised, and the viable cells were counted. The remaining three wells were stained with methyl violet (Merck, Kenilworth, NJ, USA) to count the colonies. CFU-F assay data at p0 were calculated as proportions of methyl-violet-positive colonies per total cells counted. 

The trypsinised cells after p0 were further seeded at 5000 cells/cm^2^ in low glucose DMEM (Biowest) supplemented with 10% foetal bovine serum (Gibco, Thermo Fisher Scientific), 1% glutamine, 2% penicillin, and streptomycin (all Biowest), until enough cells were obtained for the planned analyses. 

### 4.4. Multilineage Differentiation

Multilineage differentiation was performed as described previously [4,5,35]. Briefly, for osteogenesis and adipogenesis, the cells were seeded as four technical replicates in 24-well plates at 25,000 cells/cm^2^. Two replicates were used for histological assessment (one control, one treated) and two for RNA isolation (one control, one treated). The treated replicates received either osteogenic medium (growth medium supplemented with 5 mM β-glycerophosphate, 100 nM dexamethasone, 50 mg/mL ascorbic acid-2-phosphate (all Sigma)) or adipogenic medium (growth medium supplemented with 500 nM dexamethasone, 10 µM indomethacine, 50 µM iso-butylmethyl xanthine, 10 µg/mL insulin (all Sigma)). The controls received growth medium without the adipogenic or osteogenic supplements. After 21 days, the osteogenic cultures were stained with 2% Alizarin Red S, and the adipogenic cultures were stained with Oil Red O (both Sigma). After staining, the cells were imaged using Evos XL (Life Technologies, Carlsbad, CA, USA). The osteogenic potential was calculated as the concentration of Alizarin Red S (mM). The adipogenic potential was calculated as the numbers of Oil-Red-O-positive adipocytes per numbers of seeded cells, using the ImageJ software [40].

For chondrogenesis, cell pellets were formed as duplicates of 150,000 cells suspended in chondrogenic medium (high-glucose DMEM (Biowest), 100 nM dexamethasone (Sigma), 1% insulin-transferrin-selenium (Sigma-Aldrich, St. Louis, MO, USA), 50 mg/mL ascorbic acid-2-phosphate (Sigma), 1% penicillin/streptomycin (Biowest). The treated pellets received 10 ng/mL transforming growth factor ß1 (TGF-ß1; ThermoFisher Scientific), and the controls received medium without TGF-ß1. After 21 days, the pellets were fixed in 10% neutral buffered formalin (Sigma-Aldrich) and processed for paraffin sections at the Institute of Pathology, Faculty of Medicine, University of Ljubljana. The 5 µm paraffin sections were stained with Toluidine blue (Sigma) and for collagen type II (Col2) using immunofluorescence, as described previously [4,5,36]. The Toluidine-blue-stained slides were imaged using Evos XL (Life Technologies) and analyzed according to the Bern score [29]. The chondrogenic cell pellet diameters were measured using the ImageJ software [40]. The Col2 stained slides were imaged using Evos FL (Life Technologies). 

### 4.5. Immunophenotyping

The immunophenotyping was performed as described previously [4,5,35]. Briefly, culture-expanded cells between passages 1 and 5 (p1–p5) were immunophenotyped using anti-CD45 (clone 2D1), anti-CD19 (clone SJ25C1), and anti-CD14 (clone 61D3) antibodies (all ThermoFisher Scientific), as well as using anti-CD105 (clone MEM-226; ThermoFisher Scientific), anti-CD90 (clone DG3), and anti-CD73 (clone AD2) antibodies (both Miltenyi Biotec). The fixable viability dye eFluor 780 (ThermoFisher Scientific) was used to determine cell viability. Immunophenotyping was performed using Attune NTx (ThermoFisher Scientific).

### 4.6. RNA Isolation and Gene Expression Profiling

Culture-expanded cells (between p1 and p3) were used for RNA isolation and gene expression measurement of SSC markers. RNA was also isolated from cell replicates subjected to osteogenesis and adipogenesis for 21 days, as described above. In both cases, total RNA was extracted using qGOLD Total RNA kits (VWR), and the cDNA was synthesised using High-Capacity cDNA Reverse Transcription kits (ThermoFisher Scientific, Waltham, MA, USA).

Gene expression measurements were performed according to the MIQE guidelines [41]. Quantitative polymerase chain reaction (qPCR) was performed as described previously [4,5,35]. Briefly, 5× HOT FIREPol EvaGreen qPCR Supermix (Solis BioDyne OÜ, Tartu, Estonia) was used according to the manufacturer protocol. The sequences of the primers (Macrogen, Seoul, South Korea, Sigma-Aldrich) used to measure osteogenesis and adipogenesis-related genes were provided in our previous studies [4,5,35]. The sequences for the genes that encode the SSC makers were obtained from previous studies (*PDPN* [42], *CD73* [43], *CD164* [44], *CD146* [43]). All of the data were normalised to glyceraldehyde-3-phosphate dehydrogenase (*GAPDH*). 

### 4.7. Statistical Analysis

To compare the data between the three tissue groups, one-way ANOVA with Bonferroni corrections for multiple testing was used. To compare the data between the three time post mortem groups within the same tissue, two-way ANOVA with Bonferroni corrections for multiple testing was used. The statistical analyses were performed with Graph Pad Prism v8.4.3 for Windows (GraphPad Software, San Diego, CA, USA, www.graphpad.com, last access 10 February 2022). *p* values < 0.05 were considered as statistically significant. Heat maps were generated as described previously [35] using the online Heatmapper software [45]. The Figures were created using Mind the Graph (www.mindthegraph.com, last access 10 December 2021).

## 5. Conclusions

To summarise, the present study shows that BM, synovium, and periosteum harvested post mortem from the knee and peri-knee tissues retain cells with MSC-like in vitro properties and SSC gene expression profiles. Given that the cells with such characteristics can be isolated and culture expanded even >3 days post mortem, these data underpin the usefulness of the readily available post mortem tissues as a valuable source of primary cells for the purpose of regenerative medicine. 

## Figures and Tables

**Figure 1 ijms-23-03170-f001:**
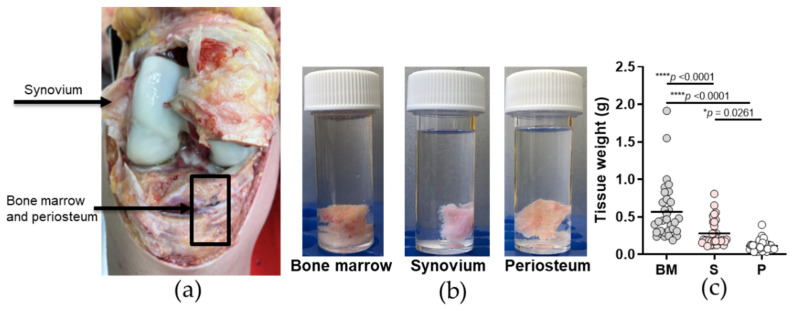
Harvesting of the knee and peri-knee tissues from the donors. (**a**) Anatomical sites for the tissue harvesting, as indicated. (**b**) Representative biopsies for each tissue, as used for primary cell isolation. (**c**) Distributions of the weights of the biopsies for each tissue group, as used for the primary cell isolation (*N* = 96 biopsies). Individual samples and means are shown, with significance where indicated (one-way ANOVA with Bonferroni multiple comparison tests). BM, bone and bone marrow; S, synovium; P, periosteum.

**Figure 2 ijms-23-03170-f002:**
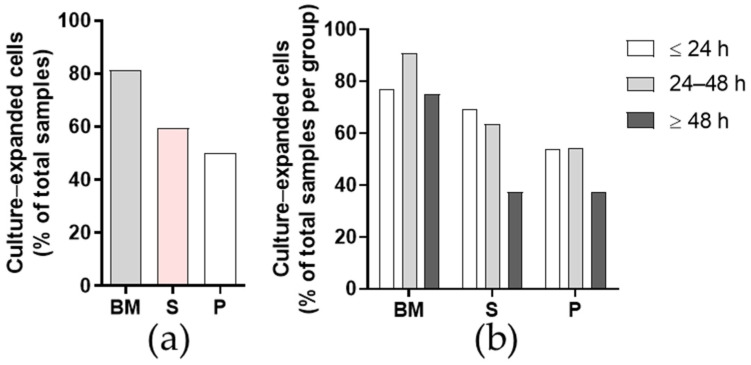
Isolation and culture expansion efficiency for the primary cells. Isolation efficiencies for each tissue (**a**) (*n* = 32 donors) and according to time post mortem groups (**b**) (≤24 h, *n* = 13; 24–48 h, *n* = 11; ≥48 h, *n* = 8). BM, bone and bone marrow; S, synovium; P, periosteum.

**Figure 3 ijms-23-03170-f003:**
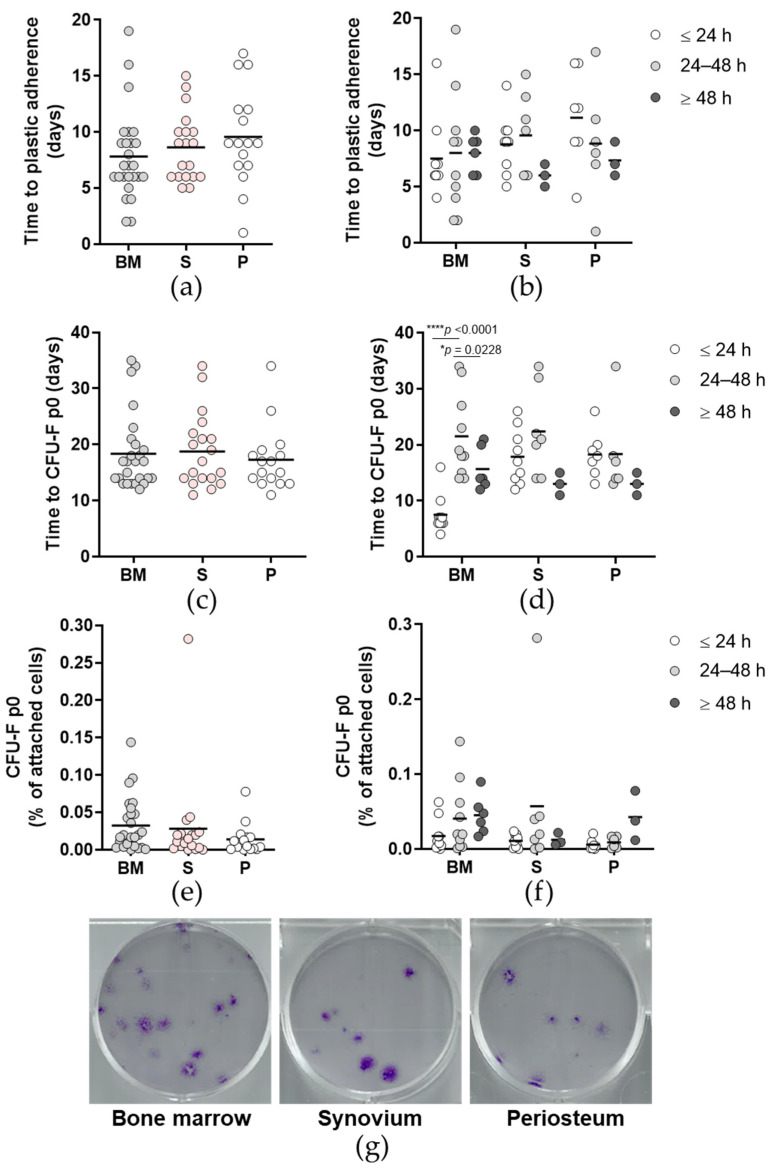
Plastic adherence and colony formation of the primary cells. (**a**,**b**) Time until plastic adherence of the isolated cells for each tissue (**a**) and according to time post mortem groups (**b**). (**c**,**d**) Time until colony forming unit fibroblast assay (CFU-F) at p0 showed no significant differences for each tissue (**c**) and according to time post mortem groups (**d**). (**e**,**f**) CFU-F activity of freshly seeded cells at p0 for each tissue (**e**) and according to time post mortem groups (**f**). (**a**–**f**) Individual samples and means are shown, with significance where indicated (one-way (**a**,**c**,**e**) or two-way (**b**,**d**,**f**) ANOVA with Bonferroni multiple comparison tests). BM, bone and bone marrow; S, synovium; P, periosteum. (**g**) Representative images of the wells for each tissue (as indicated) stained with methyl violet for CFU-F p0 quantification.

**Figure 4 ijms-23-03170-f004:**
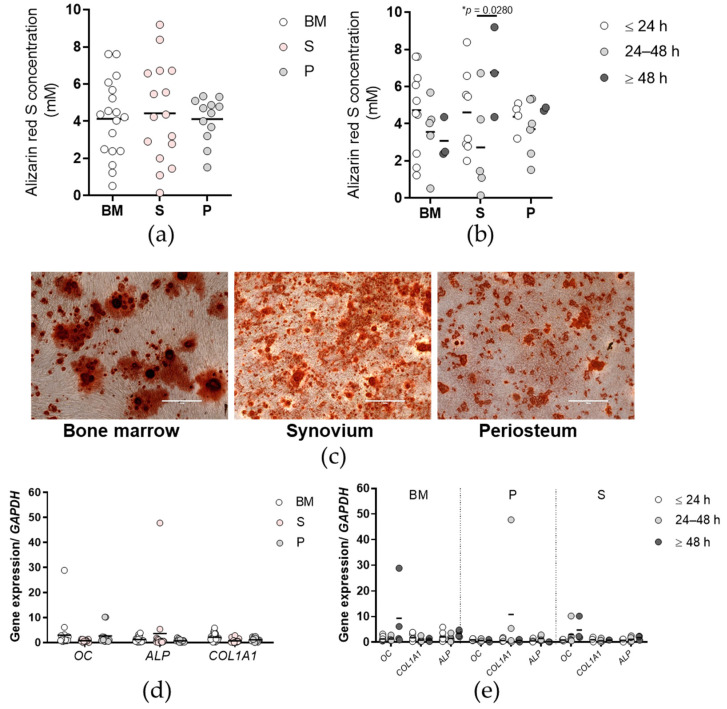
Osteogenic potential of the primary cells. (**a**,**b**) Alizarin red S concentrations for each tissue (**a**) and according to time post mortem groups (**b**). (**c**) Representative images of the wells for each tissue (as indicated) stained with Alizarin red S for rate of osteogenesis. Scale bars, 400 µm. (**d**,**e**) Expression of selected osteogenesis-related genes for each tissue (**d**) and according to time post mortem groups (**e**), as normalized to the reference gene glyceraldehyde-3-phosphate dehydrogenase (GAPDH). OC, osteocalcin; ALP, alkaline phosphatase; COL1A1, collagen type I. (**a**,**b**,**d**,**e**) Individual samples and means are shown, with significance where indicated (one-way (**a**,**d**) or two-way (**b**,**e**) ANOVA with Bonferroni multiple comparison tests). BM, bone and bone marrow; S, synovium; P, periosteum.

**Figure 5 ijms-23-03170-f005:**
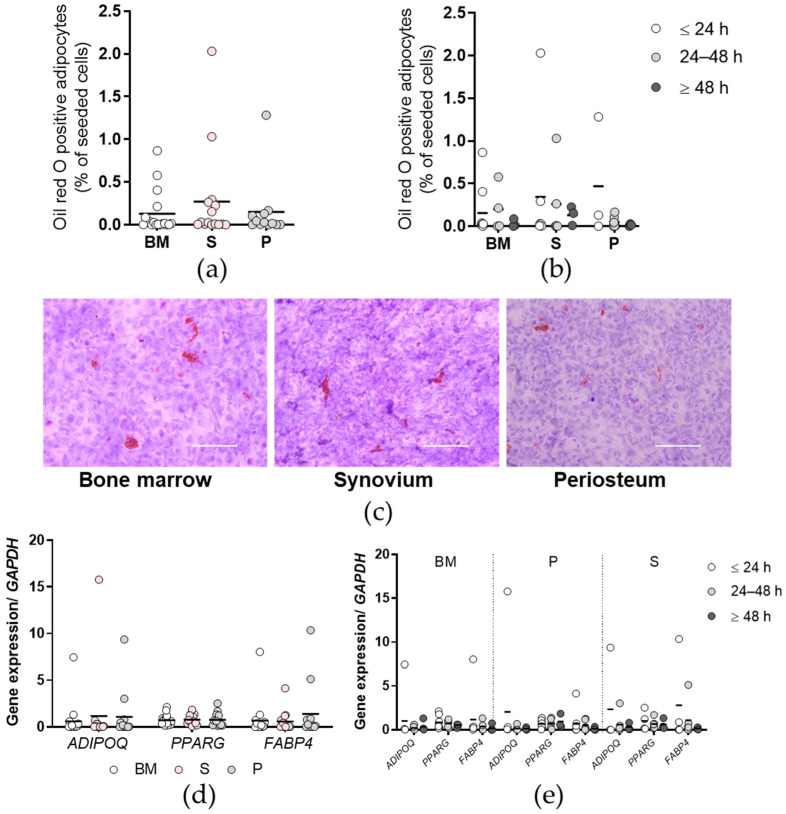
Adipogenic potential of the primary cells. (**a**,**b**) Proportions of Oil red O positive adipocytes for each tissue (**a**) and according to time post mortem groups (**b**). (**c**) Representative images of the wells for each tissue (as indicated) stained with Oil red O for rate of adipogenesis. Scale bars, 200 µm. (**d**,**e**) Expression of selected adipogenesis-related genes for each tissue (**d**) and according to time post mortem groups (**e**), as normalized to the reference gene glyceraldehyde-3-phosphate dehydrogenase (GAPDH). ADIPOQ, adiponectin; PPARG, peroxisome proliferator activated receptor γ; FABP4, fatty acid-binding protein 4. (**a**,**b**,**d**,**e**) Individual samples and means are shown, with significance where indicated (one-way (**a**,**d**) or two-way (**b**,**e**) ANOVA with Bonferroni multiple comparison tests). BM, bone and bone marrow; S, synovium; P, periosteum.

**Figure 6 ijms-23-03170-f006:**
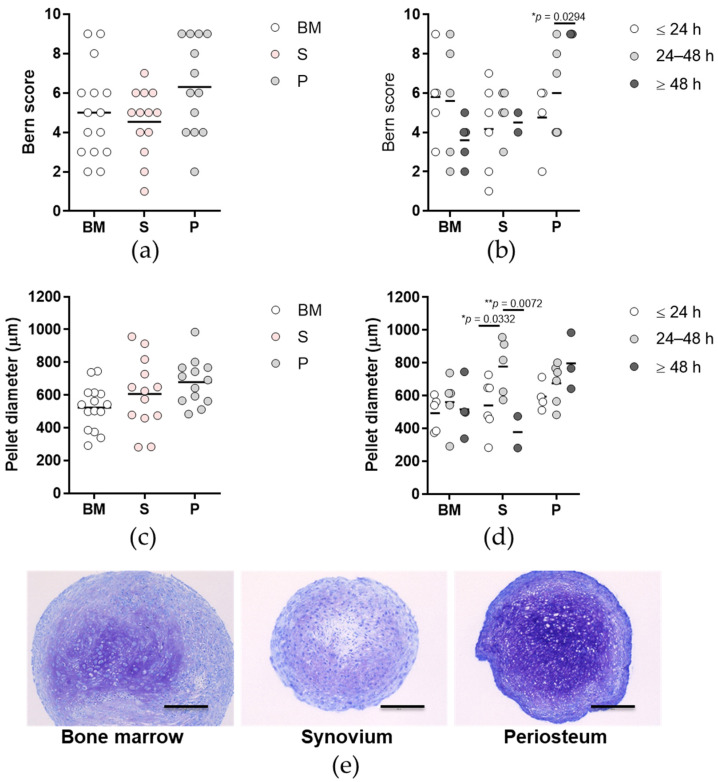
Chondrogenic potential of the primary cells. (**a**,**b**) Bern scores determined from the Toluidine blue histology for each tissue (**a**) and according to time post mortem groups (**b**). (**c**,**d**) Pellet diameters of chondrogenic cells formed by the primary cells for each tissue (**c**) and according to time post mortem groups (**d**). (**a**–**d**) Individual samples and means are shown, with significance where indicated (one-way (**a**,**c**) or two-way (**b**,**d**) ANOVA with Bonferroni multiple comparison tests). BM, bone and bone marrow; S, synovium; P, periosteum. (**e**) Representative images of the chondrogenic pellets for each tissue (as indicated) stained with Toluidine blue to determine the Bern scores and to measure the diameters. Scale bars, 200 µm.

**Figure 7 ijms-23-03170-f007:**
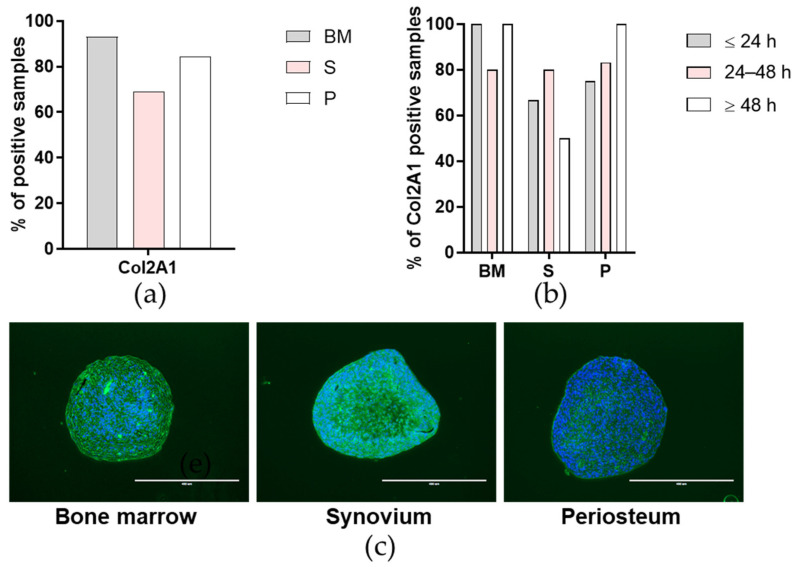
Chondrogenic potential of the primary cells. (**a**,**b**) Proportion of positive samples for α-1 chain of type II collagen (Col2A1) immunofluorescence staining for each tissue (**a**) and according to time post mortem groups (**b**). BM, bone and bone marrow; S, synovium; P, periosteum. (**c**) Representative images for each tissue (as indicated) for Col2A1 immunofluorescence (green) and 4′,6-diamidino-2-phenylindole (DAPI) nucleus staining (blue) for frequency of Col2A1 positivity. Scale bars, 400 µm.

**Figure 8 ijms-23-03170-f008:**
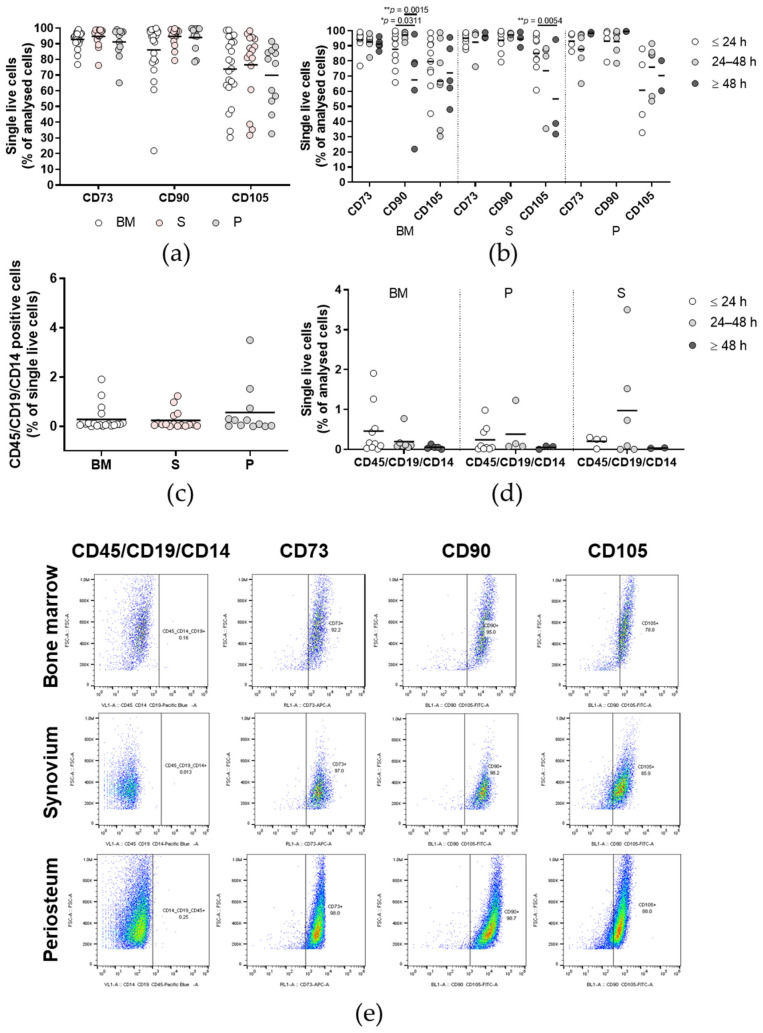
Immunophenotyping of the primary cells. (**a**,**b**) Expression of the positive markers CD73, CD90, and CD105 for each tissue (**a**) and according to time post mortem groups (**b**). (**c**,**d**) Expression of the combination of the negative markers CD45/CD19/CD14 for each tissue (**c**) and according to time post mortem groups (**d**). (**a**–**d**) Individual samples and means are shown, with significance where indicated (one-way (**a**,**c**) or two-way (**b**,**d**) ANOVA with Bonferroni multiple comparison tests). BM, bone and bone marrow; S, synovium; P, periosteum. (**e**) Representative dot plots for each tissue (as indicated).

**Figure 9 ijms-23-03170-f009:**
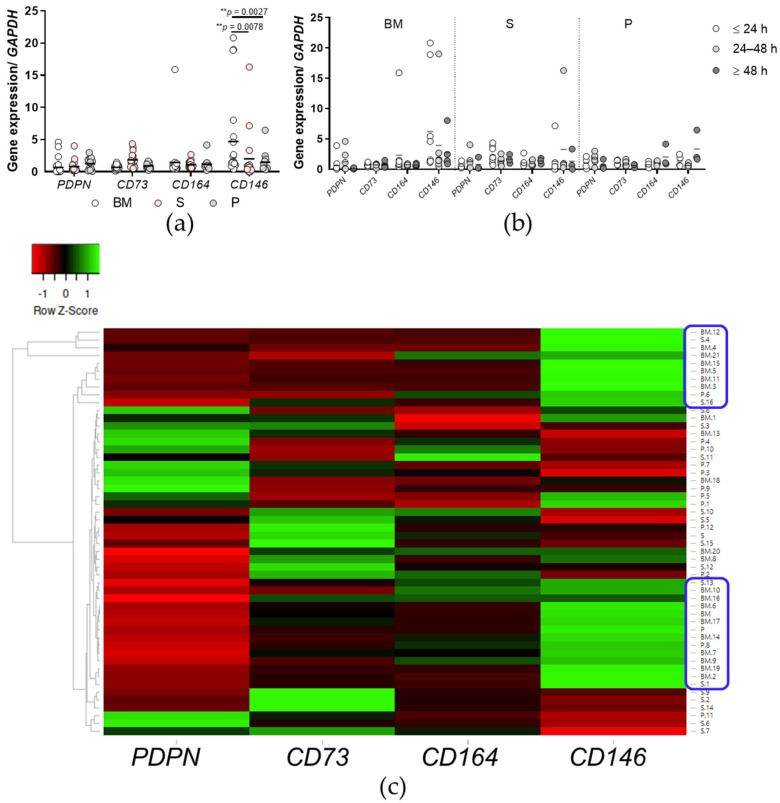
Gene expression profiling for skeletal stem cell markers for the primary cells. (**a**,**b**) Expression of the skeletal stem cell marker genes PDPN, CD73, CD164, and CD146 for each tissue (**a**) and according to time post mortem groups (**b**). (**a**,**b**) Individual samples and means are shown, with significance where indicated (one-way (**a**) or two-way (**b**) ANOVA with Bonferroni multiple comparison tests). (**c**) Heat map analysis for hierarchical clustering of skeletal stem cell marker gene expression (columns, as indicated) in the primary cells from the three tissues (rows). Green, gene expression higher than reference channel; red, gene expression lower than reference channel. Two clusters identified are shown (right; boxed in blue), along with the clustering tree analysis (left). BM, bone and bone marrow; S, synovium; P, periosteum.

**Figure 10 ijms-23-03170-f010:**
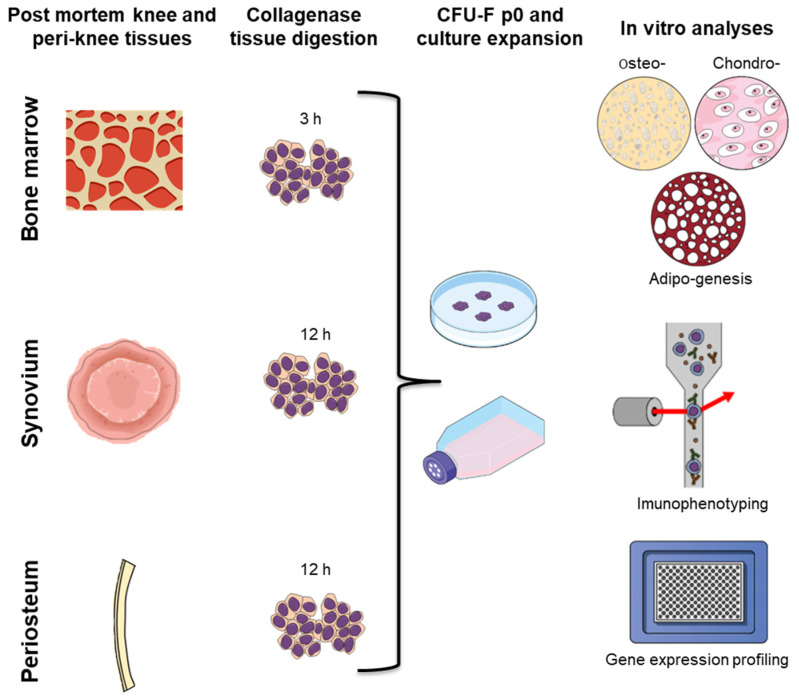
Scheme of the primary cell isolation and analyses performed in this study. CFU-F p0, colony forming unit fibroblast assay at passage 0.

**Table 1 ijms-23-03170-t001:** Donor basic characteristics and causes of death, and times post mortem of tissue collection.

Donor Number	Age (Years)	Male/Female	BMI (kg/m^2^)	Cause of Death	Time Post Mortem (h)
SM22	68	M	22.0	Suicide, hanging	24
SM23	61	F	20.8	Myocardial infarction	17
SM24	57	M	26.0	Aortic rupture	24
SM25	40	M	26.6	CO_2_ asphyxiation	39
SM26	65	F	30.1	Sudden cardiac death	4
SM27	76	F	30.1	Sudden cardiac death	23
SM28	28	M	23.6	Car accident, internal exsanguination	32
SM29	35	F	31.9	Tramadol and zolpidem intoxication	45
SM30	32	M	22.7	Ethanol intoxication	33
SM31	23	M	29.0	Suicide, hanging	33
SM32	23	M	24.6	Suicide, hanging	34
SM33	51	M	43.3	Sudden cardiac death (high BMI)	37
SM34	71	M	25.9	Acute myocardial infarction	25
SM35	77	F	27.3	Sudden cardiac death	29
SM36	57	M	24.7	Suicide, cubital incisions	24
SM37	57	F	23.5	Pulmonary embolism	5
SM38	57	M	25.9	Sudden cardiac death	101
SM39	46	F	28.0	Sudden cardiac death	69
SM40	42	M	28.7	Suicide, polytrauma	52
SM41	39	M	24.4	Suicide, shooting	14
SM42	27	M	22.6	Motorcycle accident, brain stem laceration	44
SM43	32	M	29.3	Intoxication with hypnotics and opioids	52
SM44	41	M	23.1	Sudden cardiac death	30
SM45	54	M	33.0	Car accident, chest injury	60
SM46	57	M	26.6	Epileptic attack	57
SM47	30	F	19.1	Suicide, hanging	64
SM48	44	M	31.2	Car accident, polytrauma	15
SM49	46	M	36.2	Work accident, head injury	23
SM50	61	F	23.9	Suicide, hanging	15
SM51	40	M	36.2	Methadone and bromazepam intoxication	108
SM52	51	M	31.5	Suicide, hanging	13
SM53	33	M	36.8	Suicide, shooting	9

BMI, body mass index.

**Table 2 ijms-23-03170-t002:** Mean donor characteristics per time post mortem group.

Time Post Mortem Group (h)	*N*	Age (Years)	Male/Female	BMI (kg/m^2^)	Mean Time Post Mortem (h)
≤24	13	55	8/5	27.8	16
24–48	11	41	9/2	26.7	35
≥48	8	45	6/2	28.4	70

BMI, body mass index. Except for mean time post mortem, there were no significant differences for any of these characteristics between the three groups (*p* > 0.05; one-way ANOVA with Bonferroni multiple comparison tests for age and BMI; Chi-squared tests for M/F ratio).

**Table 3 ijms-23-03170-t003:** Proportions of successfully culture-expanded primary cells per tissue according to the time post mortem groups and the number of samples for in vitro analyses carried out.

Tissue	Time Post Mortem Group (h)	Proportion of Cells/ Donors (%)	Samples for Analysis (n)					
			CFU-F p0	Osteo-Genesis	Adipo-Genesis	Chondro-Genesis	Immuno-Phenotyping	Gene Profiling
Bone/bone	≤24	76.9	10	10	9	5	10	10
marrow	24–48	90.9	10	5	4	5	7	7
	≥48	75.0	6	3	5	5	5	5
Synovium	≤24	69.2	9	8	7	6	8	8
	24–48	63.6	7	5	5	5	5	6
	≥48	37.5	3	3	3	2	3	3
Periosteum	≤24	53.8	7	4	3	4	4	4
	24–48	54.5	6	6	6	6	6	6
	≥48	37.5	3	2	3	3	2	3

CFU-F p0, colony forming unit fibroblast assay at passage 0.

## Data Availability

The data presented in this study are available on request from the corresponding author. The data are not publicly available due to restrictions such as donor privacy protection and ethical considerations.

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
