# Peer review of "Knee and Peri-Knee Tissues of Post Mortem Donors Are Strategic Sources of Mesenchymal Stem/Stromal Cells for Regenerative Procedures"

_ijms, 2022, doi:10.3390/ijms23063170_

Round 1

Reviewer 1 Report

General Comment:

This study aims to compare the colony formation, trilineage differentiation (osteogenesis, adipogenesis and chondrogenesis), immunophenotyping of mesenchymal stem/stromal cells isolated from synovium, bone marrow and periosteum from post-mortem donors. The skeletal stem cell marker-gene expression profiling in mesenchymal stem/stromal cells from these three different tissues were also evaluated in this study. Overall, this article is interesting, but I have suggestions about this manuscript:

  1. Since the authors compare the properties of mesenchymal stem/stromal cells isolated from Knee and Peri-Knee Tissues of Post-Mortem Donors with SSCs, the difference between these MSCs and SSCs should be discussed.
  2. SSCs are identified as cells that generates progenitors of bone, cartilage, and stroma, but not fat. Moreover, SSCs are present in fetal and adult bones and can also be derived from BMP2-treated human adipose stroma (B-HAS) and induced pluripotent stem cells (iPSCs). However, the authors test the trilineage potential (osteogenesis, chondrogenesis and adipogenesis) of MSCs in this study, and the gene expression profiling was tested. This should be explained or discussed.

Specific Comments:

Title:

No specific comment.

Abstract:

Line 22-24: The sentence of “Post-mortem time defined the success rate of the isolation of these primary cells.” is unprecise. I cannot understand this sentence.

Introduction:

  1. SSCs are identified as cells that generates progenitors of bone, cartilage, and stroma, but not fat. However, the MSCs are different from SSCs that MSCs are can generate not only bone and cartilage, but also as generate fat. Why authors talk about the SSCs in this introduction?
  2. Line 78: Do “Bone-marrow (BM) cells” indicate bone marrow mesenchymal stem cells or bone marrow stromal cells?

Results:

Table 1.: Does ”brain steam laceration” indicate “brain stem laceration” in SM 42 ?

Discussion:

More detailed discussion about SSC and MSCs should added. For example, the conclusion/suggestion about further application of primarily cells from synovium and periosteum should not be barely based on the gene expression profiling for SSC markers (Line 382-389).  

Materials and Methods:

No specific comments.

Reviewer 2 Report

The authors have submitted “Knee and Peri-Knee Tissues of Post-Mortem Donors are Promising Sources of Mesenchymal Stem/Stromal Cells with Favourable in-vitro Properties for Regenerative Medicine, even 3 Days Post-Mortem”.

The authors need to perform the following changes:

  1. Title must be changed to “Knee and Peri-Knee Tissues from Post-Mortem Donors are Strategic Sources of Mesenchymal Stem/Stromal Cells for Regenerative Procedures”.
  2. Currently, a growing interest has been paid on the role of extracellular environment and signalling in “regenerative medicine”; specifically, the research groups are interested on nanomedicine and exosomes: please discuss about it (See: Codispoti, B., Marrelli, M., Paduano, F., Tatullo, M. (2018). NANOmetric BIO-Banked MSC-Derived Exosome (NANOBIOME) as a Novel Approach to Regenerative Medicine. Journal of clinical medicine, 7(10), 357.)
  3. Authors have pushed their work on the role of tissue regeneration; in this landscape, a limitative role may be played by inflammation and its triggering factors (see and discuss: Bressan, E., Ferroni, L., Gardin, C., Bellin, G., Sbricoli, L., Sivolella, S., Brunello, G., Schwartz-Arad, D., Mijiritsky, E., Penarrocha, M., Penarrocha, D., Taccioli, C., Tatullo, M., Piattelli, A., Zavan, B. (2019).

Minor suggestions:

  1. Following tissue exposure to damage or other conditions such as hypoxia, metabolic acidosis occur, shifting extracellular pH. On this basis, several authors have correctly discussed on the role of acidosis on bone: here such topic has been not properly reported.
  2. The role of collagen should be here better reported (San Antonio, J.D.; Jacenko, O.; Fertala, A.; Orgel, J.P.R.O. Collagen Structure-Function Mapping Informs Applications for Regenerative Medicine. Bioengineering 2021, 8, 3.)

Round 2

Reviewer 1 Report

All my questions have been answered, and this manuscript is acceptable. I suggest accept in this present form.

Reviewer 2 Report

no comment